# Multifunctional Biomimetic Nanofibrous Scaffold Loaded with Asiaticoside for Rapid Diabetic Wound Healing

**DOI:** 10.3390/pharmaceutics14020273

**Published:** 2022-01-24

**Authors:** Sneha Anand, Paruvathanahalli Siddalingam Rajinikanth, Dilip Kumar Arya, Prashant Pandey, Ravi K. Gupta, Ruchi Sankhwar, Kumarappan Chidambaram

**Affiliations:** 1Department of Pharmaceutical Sciences, Babasaheb Bhimrao Ambedkar University, Lucknow 226025, India; snehaanand.rs@bbau.ac.in (S.A.); dkarya.rs@bbau.ac.in (D.K.A.); prashant.p@bbau.ac.in (P.P.); 2Department of Environmental Microbiology, Babasaheb Bhimrao Ambedkar University, Lucknow 226025, India; ravi.gu81@bbau.ac.in (R.K.G.); ruchi.rs@email.bbau.ac.in (R.S.); 3Department of Pharmacology & Toxicology, King Khalid University, Abha 62529, Saudi Arabia; kumarappan@kku.edu.sa

**Keywords:** nanofibers, diabetic wound healing, asiaticoside, electrospinning, extracellular matrix, crosslinking, polyvinyl alcohol, sodium alginate

## Abstract

Diabetes mellitus is a chronic disease with a high mortality rate and many complications. A non-healing diabetic foot ulcer (DFU) is one the most serious complications, leading to lower-extremity amputation in 15% of diabetic patients. Nanofibers are emerging as versatile wound dressing due to their unique wound healing properties, such as a high surface area to volume ratio, porosity, and ability to maintain a moist wound environment capable of delivering sustained drug release and oxygen supply to a wound. The present study was aimed to prepare and evaluate a polyvinyl alcohol (PVA)–sodium alginate (SA)–silk fibroin (SF)-based multifunctional nanofibrous scaffold loaded with asiaticoside (AT) in diabetic rats. The SEM findings showed that fibers’ diameters ranged from 100–200 nm, and tensile strengths ranged from 12.41–16.80 MPa. The crosslinked nanofibers were sustained AT over an extended period. The MTT and scratch assay on HaCat cells confirmed low cytotoxicity and significant cell migration, respectively. Antimicrobial tests revealed an excellent anti-microbial efficacy against *P. aeruginosa* and *S. aureus* bacteria. In-vivo study demonstrated better wound healing efficacy in diabetic rats. In addition, the histopathological studies showed its ability to restore the normal structure of the skin. The present study concluded that developed multifunctional nanofibers have a great potential for diabetic wound healing applications.

## 1. Introduction

The nanoscale world includes a diverse spectrum of materials with distinct physical and chemical properties. Currently, electrospinning has been gaining popularity due to an increasing interest in nanofibers (NF) [1]. Nanofiber is one of the best nanomaterials and is made from various polymers having a wide range of properties and applications [2]. Nanofibers have been getting much attention in recent decades, because they offer the potential to improve therapeutic benefits with lowering toxicity compared to conventional dosage forms. The characteristics of electrospun fibers, such as a large surface area, ability to load large amounts of the drug, simultaneous administration of multiple drugs and bio-active molecules, easy operation, and cost-effectiveness, have prompted the scientific community to expand in the area of improving current drug delivery systems in the last few years.

Diabetes mellitus is a metabolic condition defined by excessive glucose levels in the blood [3]. This disease significantly impacts human life due to various problems like uncontrolled blood sugar, chronic sores, vascular damage, and neuropathy [4]. Diabetes is associated with a high likelihood of wound recurrence and chronic wounds, taking years to cure. Ten to fifteen percent of the diabetic population is resistant to conventional wound healing techniques and suffers from chronic non-healing wounds regularly [5]. Chronic diabetic wounds are non-healing and persistent, which necessitates re-hospitalization [6]. Prolonged inflammation, reduced neovascularization, decreased collagen synthesis, elevated protease levels, and impaired macrophage activity are all symptoms of diabetic wounds. Furthermore, due to decreased granulocytic function and chemotaxis, diabetic wounds are more susceptible to infection [7]. The delayed wound healing in diabetic foot disease is also attributed to a variety of factors, including peripheral arterial disease, peripheral neuropathy, foot deformity, and bacterial infection. Furthermore, the wound microenvironment in diabetes is abnormal and pathogenic factors lead to delayed ulcer closure, and suboptimal volume of granulation tissue formation with abnormal extra-cellular matrix (ECM) composition [4,5,6]. An ideal wound dressing should provide a moist wound environment, offer protection from secondary infections, remove wound exudate, control bio-film, and promote tissue regeneration [6]. However, no existing dressing fulfills all the requirements associated with DFU treatment. Therefore, there is an urgent necessity for developing new, cost effective, and better remedies for diabetic foot ulcers with greater efficacy and safety in order to face this challenging situation [8].

Asiaticoside (AT) is a trisaccharide triterpene and an herbal drug (an active ingredient of Centella asiatica) which has exhibited many therapeutic activities, including anti-oxidant, anti-inflammatory, and diabetic wound-healing potential. AT benefit is limited due to the delivery problems associated with poor solubility and bioavailability [9,10,11]. Drug delivery systems and biological agents are used in bioactive dressings to promote cellular responses throughout the process of wound healing [12]. With the introduction of new biopolymers and various fabrication techniques, wound dressing materials are expected to have exceptional qualities that aid wound healing process [13]. Recently, nanofiber-based wound dressings have shown a promising potential to achieve rapid and complete healing of chronic diabetic wounds, affording significantly better results over conventional dressings owing to their unique properties, such as large surface area, nanoscale structure, providing ECM, removing wound exudates, maintaining porosity, and promoting tissue regeneration.

These properties play important roles in cell attachment and proliferation. Several novel wound dressings, including microsponge and polymeric nanofiber-scaffold, have been developed for diabetic wound healing, and they proved to be significant improvement on wound healing in diabetic-induced animal model [10,11,12,13,14].

Many researchers have reported that biopolymers like silk, chitosan, PVA, and alginate have frequently been used for wound dressing due to their inherent properties including oxygen permeability, bio-adhesion, biocompatibility, biodegradation, angiogenesis, tissue regeneration, and excellent barrier properties. For example, Xuqiang Nie et al. (2020) prepared asiaticoside-loaded nitric oxide gel for diabetic cutaneous ulcers [15]. In 2016, Lifei Zhu et al. prepared asiaticoside-loaded coaxially electrospinning nanofibers and demonstrated their excellent wound healing effect on burn injuries [16]. However, the fabrication of electrospun mat using synthetic and natural polymeric blends and an encapsulation of the active ingredient is less explored for wound healing activities. In this study, a unique blend of PCL, PVA, and SF had been prepared with the active ingredient AT. In this study we have chosen one hydrophobic polymer (PCL) and one hydrophilic polymer (PVA) to blend with silk fibroin in order to get the suitable mechanical and sustained release properties of nanofibers scaffold for effective wound healing. The formulation was further electrospun to form a mat of nanofiber. The unique mat containing curcumin-encapsulated nanofibers of natural and synthetic polymeric blend was further studied in-vivo for wound healing in streptozotocin-induced diabetic albino rats. The combinatorial fabrication of electrospun nanofibers using both synthetic and natural polymeric blends has been reported to improve both the mechanical properties and drug releasing characteristics of nanofiber in wound application [17] Therefore, we used a unique blend of PVA, SA, and SF in different combinations in order to improve the mechanical and drug release properties of nanofibrous scaffold. Furthermore, the nanofibrous scaffolds were cross-linked with glutaraldehyde as a cross-linker to achieve the mechanical and sustained release properties required for efficient wound healing. The unique multifunctional nanofibrous scaffold loaded with AT was further investigated for its in vitro characteristics and in vivo wound healing efficacy in streptozotocin induced diabetic rat model.

## 2. Materials and Methods

### 2.1. Materials

Polyvinyl alcohol (PVA) and sodium alginate (SA) were purchased from Himedia Laboratories Pvt. Ltd. (Mumbai, India). Asiaticoside (AT) drug was purchased from Shaanxi Pioneer Biotech Co. Ltd. (Shaanxi, China). Dimethyl sulfoxide (DMSO) was obtained from S D Fine Chem. Ltd. (Mumbai, India). Streptozotocin (STZ) was bought from Sigma Aldrich (Bangalore, India). Formic acid was obtained from Avra Synthesis (Hyderabad, India). Ketamine was purchased from Miracalus Pharma Pvt. Ltd. (Mumbai, India). Silkworm cocoons were obtained as a gift sample from the Department of Zoology, Babasaheb Bhimrao Ambedkar University (Lucknow, India). All other used chemicals or reagents were of analytical grade.

### 2.2. Preparation of Electrospinning Solutions

PVA (8% *w*/*v*) and SA (2% *w*/*v*) were soaked overnight and magnetically stirred continuously in distilled water for 6 h at 40 °C to obtain a uniform solution. The obtained silk cocoons were first degummed two times at 100 °C for 30 min using 0.1 molar Na_2_CO_3_ and 0.5 percent as a standard detergent. After they were degummed, the cocoons were washed twice with warm water to remove all the silk sericin from its surface and air dried it. To fully dissolve the silk fibroin (SF), formic acid and calcium chloride were added in the ratio of (1:0.25 *w*/*w*). To create a transparent silk fibroin solution, the resulting solution was stirred overnight [18]. The prepared SF solution and the polymeric blend (PVA & SA) were blended in a 7:3:2 ratio, then mixed vigorously with a high-speed homogenizer. To ensure thorough dissolution and homogenous solutions, the final combinations were agitated at room temperature for 2 h. To create a homogenous viscous solution, AT was added to the final mixture while stirring continuously and vortexed for 10 min. To obtain AT loaded PVA/SA/SF solution, the resulting solution was agitated for 24 h at room temperature.

### 2.3. Fabrication of AT Loaded Electrospun Nanofibers

The stable and homogenous AT-loaded solutions were electrospun using electrospinning (ES1, E-Spin Nanotech Pvt. Ltd., Kanpur, India). The electrospin parameters like applied voltage, needle-to-collector distance, and solution flow rate, were tuned before initiating the electrospinning. Furthermore, the 10 mL prepared solution was placed into a syringe with a 0.4 mm inner diameter stainless-steel flat needle tip. The high-voltage power supply was provided by connecting the positive electrode to the needle. Electrospinning was carried out with a distance of 10 cm between the syringe pump and a round cylindrical collector wrapped with aluminum foil. The rotating drum collector speed was kept at 300 rpm. The solutions were pumped at a flow rate adjusted between 15–25 kV depending on the formation of the Taylor cone once. The whole process was carried out under a defined humidity percentage. 

### 2.4. Crosslinking of Nanofibers

The crosslinking (CL) was done by incubating AT-loaded nanofibrous scaffold in desiccators saturated with the glutaraldehyde vapor 50 percent *v*/*v* aqueous solution for 2 h; the AT-loaded nanofibrous scaffold was cross-linked with vapors of glutaraldehyde. Following cross-linking, samples were placed in a vacuum oven at room temperature for 24 h at 110 °C to remove glutaraldehyde which was unreacted. The samples were preserved in another desiccator until they could be used again. Cross-linking was assessed by weighing specimens of nanofibrous scaffold and determining the degree of swelling and nanofibers weight loss.

### 2.5. Surface Morphology of AT Loaded Electrospun Nanofibers 

The morphology of the electrospun AT-loaded SF with both PVA and SA nanofibers (AT-PVA-SA-SF) surface was evaluated using SEM (scanning electron microscopy). A scanning electron microscope (JSM-649OLV, JEOL, Tokyo, Japan) was used at an accelerating voltage with various magnifications. To obtain electrically conductive surfaces, the nanofiber samples were mounted on SEM specimen stubs and sputter-coated with Au-Pd in an argon environment for 1.5 min. Image J software (National Institutes of Health and the Laboratory for Optical and Computational, Madison, WI, USA) was used to determine the average diameter of nanofiber scaffold. The diameters of the fibers were measured at various points [19].

### 2.6. Mechanical Strength of Nanofibers

It is crucial to determine the mechanical strength of nanofibers because during their lifetime, types of forces exerted on them, in any form, may result in any permanent or temporary deformation or even sometimes failure. Briefly, AT-loaded nanofibers were put through their paces on a tensile tester (Zwick Roell Z005, Ulm, Germany). According to ASTM D683-Type V, the nanofiber samples were chopped into dog bone shapes (10 to 30 mm). The ends of the samples were attached to the tensile testing machine’s gripping units, and a load of 10 kN was applied at a rate of 1 mm/minute until the samples broke [20].

### 2.7. Measurement of Swelling Ratio or Water Uptake 

The limit of water uptake or swelling proportion is a fundamental boundary for assessing nanofibers for wound recuperating and is usually tried by the gravimetric strategy [21]. Briefly, the AT-loaded nanofibrous membranes were cut into 1 × 1 cm, and their dry weight (W_d_) was estimated by an electronic weighing balance and noted. These weighed samples were immersed in Phosphate Buffer Solution (pH 7.4) at room temperature. At certain predetermined intervals, the films were taken out from the media and put on a piece of tissue paper to eliminate overabundance water which was clung to the surface of the nanofibers. The wet load of the nanofibers (W_w_) was estimated right away. All estimations were performed three-fold. The enlarged proportion or level of wateruptake was determined by the following Equation (1): [22]
(1)$$\%\;\mathrm{Swelling}\;\mathrm{ratio}\;=\frac{W_{w}-W_{d}}{W_{d}}\times 100$$
where W_w_ is the wet sample weight after swelling, and W_d_ is the dry weight of the sample.

### 2.8. Measurement of In Vitro Biodegradability 

The weight loss at different intervals was used to investigate the degradation behavior of nanofibers. AT-loaded nanofibers were vacuum dried for 24 h at 50 °C before the in vitro biodegradability test. In addition, the nanofibers were sliced into squares (1 × 1 cm^2^) and weighed with an electronic balance. They were put in a PBS (pH 7.4) with lysozyme to test enzymatic breakdown. Each sample was placed in a test tube containing 10 mL of PBS and 3 mg lysozyme (104 unit/mL) to replicate the physiological body environment. The nanofiber samples were placed in a shaker incubator at 37 °C and 80 revolutions per minute. To assess for weight loss in nanofibers due to enzymatic degradation, they were withdrawn from the enzyme solution, rinsed with distilled water, dried for 1.5 h, and weighed every 24 h for 14 days. The weight loss (W_L_) of the samples was calculated using the following Equation (2), and the degradation profiles were created by averaging three samples and comparing the mass at each time interval (W_t_) with the initial weight (W_i_) [23,24].
(2)$$\%\;\mathrm{Weight}\;\mathrm{Loss}\;=\frac{W_{i}-W_{t}}{W_{i}}\times 100$$

### 2.9. In Vitro Drug Release Study

The in vitro drug release characteristics of asiaticoside-loaded nanofibrous scaffold(AT-PVA-SA-SF) was determined in phosphate buffer solution (PBS) pH 7.4 ± 0.2 at 37 °C with 100 rpm in a shaking incubator. During in vitro release tests, sink conditions were maintained. Briefly, the AT-loaded nanofibers (1 cm^2^) were incubated at 37 °C for 24 h in 10 mL PBS. 1 mL sample solutions were taken from the dissolution medium at predefined time intervals (0.5, 2, 4, 6, 8, 10, and 12 h) and replaced with the same amount of new medium. The amount of asiaticoside was determined using a calibration curve and a UV spectrophotometer with a 210 nm wavelength to quantify the amount of AT released at different time intervals. The amount of drugs released was graphically plotted against time as a percentage. All measurements were made three times, and the results are presented as averages with standard deviations. Nanofiber drug loading was calculated theoretically.

### 2.10. Anti-Microbial Activity

#### 2.10.1. Formation of the Zone of Inhibition

Antibacterial activity of drug-loaded nanofibers was tested against Gram-negative and Gram-positive bacteria *Pseudomonas aeruginosa* (ATCC 27853, Boston, MA, USA) and *Staphylococcus aureus* (ATCC 25923, Seattle, WA, USA), respectively, through the agar well disc diffusion method. *Pseudomonas aeruginosa* (*P. aeruginosa*) was grown in Luria Bertani broth (LB), and *Staphylococcus aureus* (*S. aureus*) was grown in Tryptone soy broth (TSB) (Himedia LQ508) liquid media for 24 h, 37 °C at 200 rpm in an orbital shaker. The bacterial cultures were spread on a Mueller Hinton agar plate (MHA) (Himedia; M173) with the help of sterile cotton bud stick separately in triplicates. 6 mm wells were made on the MHA plate by a sterile cork borer under aseptic conditions. The drug encapsulated nanofibrous discs (5 mm) were punched and sterilized under UV light for 2 h. Drug-loaded nanofibrous disc, drug-loaded silk firoin disc, gentamycin-impregnated disc against *P. aeruginosa* and vancomycin disc against *S. aureus* were used as a positive control; a placebo disc as a negative control was kept separately in the agar well under sterile conditions. The plates were incubated at 37 °C for 24 h. The antimicrobial activity was measured by a clear zone of inhibition in mm and documented. The experiment was performed in triplicate, and the average value of clear zone inhibition was shown in the results.

#### 2.10.2. Time-Kill Assay

Time kill assay is an appropriate method to detect the time-dependent killing of bacteria in the presence of active drug or nanofibrous sheets. This method calculates the dynamic interaction between microbes and antimicrobial agents during the time of incubation [25,26]. The anti-bacterial activity of drug-loaded nanofibers against *P. aeruginosa* and *S. aureus* was determined by a time kill assay. Briefly, nanofibrous mat (5 cm), negative control (without drug), and positive control (gentamycin 0.5 μg/mL for *P. aeruginosa* and vancomycin 1.0 μg/mL for *S. aureus*) were added to 1 mL culture (*P. aeruginosa* or *S. aureus*) in separate test tubes (modified to 1 × 10^6^ CFU/mL as a starting inoculum). It is to be noted that positive control culture tubes received the antibiotics at sub-MIC concentration. All the test tubes were incubated in an orbital shaker for 24 h, 37 °C in shaking condition at 150 rpm. The CFU/mL were calculated at 0 h, 6 h and 24 h after plating of 100 µL of appropriate dilution factor on a Luria Bertani agar plate (for *P*. *aeruginosa*) and Tryptic soy agar plate (for *S. aureus*). All plates were incubated at 37 °C and CFU (colony forming unit) was counted after 24 h. Both the experiments were performed in a triplicate manner for statistically significant data between nanofibers, positive and negative control [27].

#### 2.10.3. Microbial Penetration Test

The invasion of bacteria into the wound is the first step of pathogenesis. It is interesting to check the capability of nanofibrous scaffold to avoid microbial penetration into the wound. We have evaluated the resistance of microbial invasion by nanofibrous scaffold in vitro. Three glass vials of 15 mL were filled with 5 mL of nutrient broth (NB) media; one vial was covered with AT-loaded nanofibrous sheet, one was plugged with a cotton ball and the third one was kept in an open condition. All vials were left at room temperature. After 3 days and 7 days, the bacterial growth of media containing vials were measured. Vials plugged with cotton and open served as positive and negative controls, respectively. Microbial growth was measured by the visible spectroscopy at 600 nm wavelength and colony forming unit [27]. 

### 2.11. In Vitro Cell-Line Study

#### 2.11.1. Cell Culture and Treatment

In 75 cm^2^ flasks, human keratinocytes (HaCaT, NCCS, Pune, India) were grown in full Dulbecco’s Modified Eagle’s medium (DMEM), which consisted of DMEM medium with 10% heat-inactivated fetal bovine serum and 1% L-glutamine. Detaching cells from flasks was done with an EDTA-trypsin solution, and cell counting was done with a hemocytometer and Trypan Blue staining.

#### 2.11.2. In Vitro Cytotoxicity (MTT Assay)

The MTT test is a helpful tool for deciding the cytotoxicity of biomaterials in vitro. The yellow tetrazolium salt in MTT is changed over to purple formazan within the presence of phenazine methosulfate (PMS) by the dehydrogenase proteins in live cells in this trial. The measure of formazan created is identified with the number of living cells in the framework [28]. HaCaT were seeded at a density of 1 × 10^4^ cells per well (in 100 µL of DMEM media) and cultured for 24 h in a 96-well plate. The plates were incubated for 24 h after the medium was supplanted with nanofibers. From that point onward, 10 mL of 5 mg/mL MTT reagent was added to each well and incubated for an additional 4 h. The purple formazan was solubilized by adding 100 µL dimethyl sulfoxide to the entirety of the wells, including the control (which got no treatment), twirling tenderly to blend well, and afterward keeping at room temperature for around 30 min. The absorbance at 570 nm was estimated utilizing a microplate reader [29]. 

#### 2.11.3. In Vitro Scratch (Cell Migration) Assay

The nanofibrous scaffolds loaded with AT were cultured in a DMEM medium at 37 °C. After 48 h, the release media was collected. HaCat cells were cultured in the wells of a 6-well plate at a concentration of 1 × 10^5^ cells mL^−1^, and the plates were incubated at 37 °C for 24 h to create confluent monolayers. Then, using a 200 µL pipette tip, a gap was made on the monolayer of cells, and the debris was eliminated by washing the cells two times with warm PBS. As a blank, the medium was utilized without the sample. The samples were imaged consistently at 0 and 48 h to assess the cell migration for each treated nanofiber. An optical microscope was utilized to capture pictures of the scratched regions. For each well, five pictures were taken after arbitrary choice [30,31].

### 2.12. In Vivo Studies

For in vivo studies, the approved (BBDNIIT/IAEC/2019/10/01) animals (Albino Wistar male rats) were procured from the Indian Veterinary Research Institute (IVRI) Bareilly, Uttar Pradesh, India. All animals weighing 100–150 g were selected and kept separately in polypropylene cages under pathogen-free standard temperature conditions (23 ± 2 °C) and a humidity of 55 ± 5% with 12 h light and 12-h dark cycles. A standard laboratory meal (commercial pellet diet) was employed for feeding, along with an unlimited supply of fresh drinking water. The animals were placed into five groups, each with five animals.

#### 2.12.1. Inducing Diabetes and Measuring Body Weight

All the animals of average weight 100–150 g were used for inducing diabetes. Firstly, they all fasted overnight, and then streptozotocin was administered through the intraperitoneal route with a dose of 60 mg/kg, using a 0.1 M cold citrate buffer (pH 4.5) to induce diabetes mellitus symptoms. Streptozotocin (STZ) induces diabetes within 2–3 days by destroying the beta cells. A glucometer was utilized to check expanded blood glucose levels 48 h after STZ injection. Rats with blood glucose levels higher than 130 mg/dL will be utilized in further study. All diabetic rats will be weighed to look at any deficiency of body weight because of diabetes [23,32].

#### 2.12.2. In Vivo Wound Healing Study

Animals were given anesthesia by the intraperitoneal route through injections of ketamine (80 mg/kg) and xylazine (5 mg/kg), and the hairs from the dorsal region were shaved off to obtain hairless skin in order to create wounds. A betadine solution was used to sterilize the skin around the wounds. On the back of each animal, a single circular full-thickness wound was made with a disposable 8 mm skin biopsy punch. Nanofibers were applied to wounds by putting on them in touch with subcutaneous tissue. Every nanofiber-covered injury was secured with a Tegaderm TM covering to forestall bacterial contamination. Following the treatment, the nanofibers on the injury were eliminated under anesthesia at different spans (Day 0, 3, 6, 9, and 14) and replaced with new nanofibers. Advanced photographs of the injured region were acquired at 0, 3, 6, 9, and 14 days for the actual appearance and wound conclusion. They were taken by zooming in the camera focal point upward on the injury’s middle and keeping a 6-cm separating between the camera and the injury. The decrease in injury region was utilized as a proportion of treatment viability. The decrease in injury region was depicted as far as rate decrease of wound region and determined utilizing the Equation (3) [6,33].
Rate of wound healing = (AW_i_ − AW_m_)/AW_m_ × 100(3)
where AW_i_ is the initial wound and AW_m_ is the remaining wound after nth day of treatment.

### 2.13. Histopathology

Animals from every group were euthanizeded, and wound biopsies were taken on the various days (day 0, 7 and 14) of the experiment with a 2-mm edge of wounded skin. All the samples were kept overnight in 10% buffered paraformaldehyde, followed by sequential ethanol dehydration and then embedding in paraffin with a microtome. Then the tissue was cut into ten mm thick sections and was stained with hematoxylin and eosin as per the standard protocol. The samples’ parameters were further assessed under light microscopy.

### 2.14. Statistical Analysis

The statistical analysis of all the data was performed utilizing Origin Pro 19 software (Origin Pro Lab Corporation, Northampton, MA, USA). Results were communicated as the mean ± standard deviation. Data were analyzed by one-way analysis of variance, and the differences among all the means of groups were examined by an unpaired, two-sided Student’s *t*-test. *p*-value less than 0.05 considered as significant, while *p*-value less than 0.01 was considered to be highly significant.

## 3. Results and Discussion

### 3.1. Morphology of AT-Loaded Nanofibers

The electrospinning approach was effective in creating nanofibrous scaffold. The parameters, like concentration of polymer, voltage, the distance between pump and collector, and flow rate, etc., were chosen based on research and optimization of the electrospining process. SEM assessment demonstrated the surface morphology, diameter, and homogeneity of nanofibers of the optimized nanofibrous scaffold. The morphology of only silk fibroin nanofiber was indiscriminate with the unpleasant surface, and the fibers were not very well shaped (Figure 1E). On the other hand, when SF mixed with PVA and SA at different ratio and composition, the morphology improved altogether and the consequences of the SEM picture showed fine, smooth, and interconnected all around organized nanofibrous mats with web-like permeable construction as is shown in Figure 1A,C. Furthermore, in the SEM images from Figure 1A,C, no other heterogeneous structure was observed on surface of nanofiber, indicating that the AT was dissolved completely and mixed uniformly in polymeric solution before the electrospining process.

The normal and average diameter of the non-crosslinked nanofibers was found to range from 100–140 nm, as shown in Figure 1B. The nanofibers were formed individually with uniform distance across their lengths. The diameters of the crosslinked nanofibers expanded, shown in SEM image Figure 1C and graphical representation Figure 1D, because of their response to glutaraldehyde [34]. In view of past research, the observed sizes were optimal for the multiplication of fibroblast, cell adhesion, and cell attachment. The pre-arranged nanofibers were mimicking the skin’s normal extracellular matrix (ECM), and the web-like design gives the strands a huge surface region, which is significant for cell breath, oxygen supply, and moisture content support, all of which assist with rapid wound healing. The findings showed that the developed nanofibrous scaffold exhibited all desired physical and structural properties for rapid wound healing.

### 3.2. Mechanical Strength of Nanofibers

As described in earlier research [23], the mechanical strength is measured to find out the force required to withstand the maximum stress per unit area for a sample. A tensile testing machine was utilized to decide the mechanical strength of produced nanofibers. In a pure silk fibroin nanofibrous mat, a regular fragile crack with poor rigidity was observed. When silk fibroin was combined with PVA/SA polymers, the tensile strength of the nanofiber was enhanced greatly. As silk fibroin creates bonds with other polymer molecules, numerous random coil conformations of its structure are changed into β-sheets, which may explain why silk tensile strength is increased when combined with PVA/SA as shown in Figure 2. The results from Figure 2 showed that the cross-linking of nanofibers further enhanced the mechanical strength of nanofiber up to 20.65 MPa, which is compatible with human skin [35]. The tensile strength of optimized formulations was presented in Table 1. Several authors have been investigated and reported nanofibrous formulations with tensile strengths ranging from 6 to 18 MPa are appropriate for cutaneous application [36]. The tensile strength of the developed nanofibrous formulation was in the range of 12–20.65 MPa, which is within the ideal range.

### 3.3. Swelling Ratio or Water Uptake Study

The nanofibers’ capacity of water absorption or retention was tested to determine the number of exudates that the fiber could absorb when used as a dressing. As shown, the water absorption capacity of the nanofibrous scaffold in a phosphate buffer of pH 7.4 increased over time. The AT-PVA-SA-SF water absorption capability has increased with time due to the hydrophilic nature of PVA, which allows water to diffuse into its fibers. Furthermore, the water absorption capacity of non-crosslinked nanofibers rose rapidly until 6 h, after which it remained unchanged because of lower porosity with closed pores [37]. Furthermore, the hydrophobic character of nanofibers caused by cross-linking may hinder water molecule transport in its fiber mat after 3 h, resulting in a decrease in fiber mat water uptake capacity (Figure 3).

### 3.4. In-Vitro Biodegradation Study

Nanofiber scaffolds were immersed in phosphate buffer at pH 7.4 for 14 days to test in vitro biodegradability. The non-crosslinked formulation disintegrated completely on day 8, yet the cross-linked one went up to 14 days to disintegrate completely. The distinction in nanofiber degradation time is plainly owing to the hydrophilic nature of the two polymers; the higher the hydrophilicity, the quicker the degradation as displayed in Figure 4. Because both polymers utilized in the formula were hydrophilic, they degraded within 8 days compared to a cross-linked formulation of the same polymers. On the other hand, the result clearly shows that non-crosslinked nanofiber formulations, regardless of composition, biodegrade within a few days of application while crosslinked formulation degrade slowly [37].

### 3.5. In Vitro Drug Release

In the solution of phosphate buffer dissolving of pH 7.4, in vitro drug release from produced nanofibrous formulations was measured. Due to the degrading polymeric layer of the fiber, non-crosslinked nanofibers exhibit a pattern of initial burst drug release within the first few hours due to the hydrophilicity of the polymers. Nanofiber dosage forms’ quick burst release aids in achieving the minimal effective drug concentration required to produce the desired and quick pharmacological reaction in a short period of time. Cross-linked nanofibers, on the other hand, follow a prolonged release of the active drug molecule from tight pores of the core of the fiber, maintaining the same therapeutic response for a longer period (Figure 5). This could be due to the hydrophilic nature of both polymers, as well as their high swelling rate, which allows drug release by diffusing through the swollen pores, whereas cross-linked nanofibers, due to their lower swelling rate, hold drug molecules for a longer time and can also release the drug from nanofibers at a much slower rate due to their lower water uptake capacity [38].

### 3.6. Anti-Microbial Study

#### 3.6.1. Formation of Zone of Inhibition

The study of antimicrobial activities of nanofibrous scaffold for proper wound dressing and healing is an important phenomenon. AT-loaded nanofibers protect the wound from microbial invasion and help with tissue regeneration. The antibacterial activity of AT nanofibers was determined by the modified agar well disc diffusion method against *P. aeruginosa* and *S. aureus.* A clear zone of inhibition shows the antibacterial activity of AT-loaded nanofibrous scaffolds. Agar well disc diffusion assessment against *P. aeruginosa* and *S. aureus* indicated that non-crosslinked silk-fibroin-based AT nanofibers formed a greater zone as compared to cross-linked AT nanofibrous scaffold against both bacterial strains. The results of Figure 6 suggest that the antibacterial potency enhanced by increasing the drug concentration in nanofibrous formulations.

#### 3.6.2. Time Kill Assay

Bactericidal activity of drug nanofibrous scaffolds was assessed by time-kill assay. The results suggest that AT-loaded nanofibrous scaffolds are more effective in killing the *S. aureus* (Figure 7) and *P. aeruginosa* (Figure 8) at 6 h and 24 h as compared to AT drug alone. However, AT alone was more effective against *S. aureus* as compared to *P. aeruginosa* at 6 h.

#### 3.6.3. Microbial Penetration

Protect the wound from microbial infection through proper dressing and penetration barrier. In this investigation, a cotton-plugged vial was used as a negative control that indicates there was no microbial contamination in the vial; on the other hand, an open vial showed turbidity after 3 days and 7 days, which confirmed that various types of microbes entered into the nutrient broth due to the lack of any barrier. We performed the absorbance and CFU counts of all three vials at day 0, day 3, and day 7 in triplicate. The results according to Figure 9 suggest that only the positive control vial, which remained open, had turbidity and CFU (colony forming units) counts, which suggests the protective efficacy of the NF-capped vial against the invasion of microbes. The results of microbial penetration assay are graphically represented as a bar diagram in Figure 10.

### 3.7. In Vitro Cytotoxicity (MTT Assay)

The MTT assay estimates the number of viable cells by lowering the MTs tetrazolium compound by viable active cells, which reflects normal mitochondrial function, whereas Trypan Blue is predicated on the assumption that healthy cells have intact cells membranes that are not stained. Over 24 h of cultivation, in vitro cytotoxicity tests revealed less cytotoxicity of HaCaT cells by the AT nanofibers membrane compared to placebo and control as shown in Figure 11. Furthermore, the number of cells rises constantly throughout 24 h of growth, demonstrating that all nanofibers are non-toxic, according to the optical density of the MTT assay. Cell viability of more than 80% is deemed non-toxic. These findings provide scientific proof that AT-loaded nanofibrous membrane is less-toxic to HaCaT cells. In addition, findings from an in vitro cytotoxicity assay show that AT-loaded nanofibers are biocompatible and less toxic to HaCaT cells with more significant growth rates.

### 3.8. In Vitro Scratch Assay

Re-epithelialization is a crucial cycle which forces reemerging of the injury with new epithelium for which appropriate relocation and expansion of keratinocytes at the outskirts of the injury is important [39]. To assess the impacts of each twisted dressing on keratinocyte (HaCaT) movements, the in vitro wound recuperating examination was led. Scratch wounds were made on a monolayer of cells and checked up to 24 h, then pictures were taken. As displayed in Figure 12, nanofibers worked on the capacity of HaCaT cells to move into the scratched region when contrasted and control. The distance of the holes was comparative across every one of the examples during the primary hours. After treatment, AT nanofibers F1 (non-crosslinked) and F2 (crosslinked) both fundamentally upgraded the relocation of HaCat cells toward the open region when contrasted with different gatherings. The outcomes affirmed that the nanofibers containing AT could improve keratinocyte relocation and growth in vitro, which is vital for the wound re-epithelialization process [30,31].

### 3.9. Measurement of Body Weight and Diabetes

To induce diabetes, 60 mg/kg streptozotocin was given intraperitoneally. The mean increase/decrease in body weight was measured and recorded for all animal groups at various time intervals after causing diabetes in them. After inducing diabetes in rats, the average body weight was observed for any gain/reduction in all the groups and recorded at various time intervals. The average body weight of all groups of rats decreased after diabetes was induced and before therapy, indicating that the diabetes was properly induced in all the animals. When comparing the normal control group and nanofiber-treated groups on days 0, 3, and 14, it was discovered that there was a significant (*p* = 0.01) drop in the toxic group of animals after treatment. The body weights of treatment groups, however, did not differ significantly. The weight loss and blood sugar levels of rats were tested before and after diabetes induction, and we discovered that blood sugar levels were higher (as shown in Table 2) and the weight of rats was lower (as shown in Table 3).

### 3.10. In Vivo Wound Healing Study

In streptozotocin-induced diabetic rats, the wound healing efficacy of nanofibers loaded with asiaticoside was assessed at various time intervals. For days 0, 3, 6, 9, and 14, the wounded areas were photographed from the same height or distance, and the results were reported in Figure 13. To determine the healing potential of nanofibers, the wounds were examined and the area of wound was quantified on days 0, 3, 6, 9, and 14 and represented in Figure 14. On the indicated days, the wound healing rate of the optimized nanofiber formulation batches was higher than that of the control groups of animals. The images of wounds were assessed with the software Image J at a scale bar of 4 mm to determine the wound area. Figure 13 clearly showed that on the 14th day of treatment, the AT loaded silk-based nanofibrous scaffolds had a greater wound healing efficacy than the toxic and placebo groups (*p* = 0.01). The large surface-area-to-volume ratio of nanofibers, which resulted in increased exudate absorption, was linked to the effective wound healing capability of nanofibers. Furthermore, the wound healing capabilities of natural silk fibroin mixed with the properties of asiaticoside may have improved wound healing in the animals. Furthermore, the web-like porous structure of the prepared nanofiber, which biomimetics the extracellular matrix (ECM) of the skin surface, may have allowed required cell respiration and maintained required water and oxygen permeability, allowing for a moist ambient environment at the site of the wound, which is necessary for effective healing of the wound.

### 3.11. Histopathology

The lesions were removed and histological examinations were performed a few weeks after damage. Wound tissues from different animal groups were collected at days 0, 7, and 14 (stained with Haemotoxylin & Eosin) and photographs were made for the visual evaluation of the tissue’s inner structure. On days 7 and 14, histological pictures of wound tissue treated with nanofibrous formulations revealed the presence of an intact layer of keratin with regular stratified squamous epithelium cells. As shown in Figure 15 the skin cells were constructed with more fibroblasts and less inflammatory cells, indicating that the proliferation phase had begun and healing was taking place, as demonstrated by the epithelial layer regeneration. Furthermore, connective tissue had fibrin and fibroblast in moderate amounts, indicating the synthesis of collagen. The lack of vascularisation in photos taken from nanofiber-treated groups confirmed the absence of inflammation According to the findings, the maturation and remodeling stages of wound healing were completed on day 14 in the nanofiber-treated group.

## 4. Conclusions

In diabetic patients, poor wound healing is one of the most severe consequences, leading to amputation. The electrospinning technology was used successfully to fabricate AT-loaded silk fibroin multifunctional nanofibrous scaffold mixed with various compositions of PVA and SA. The nanofibers displayed the critical physicochemical characteristics required for cutaneous wound healing applications. The in-vivo investigations on diabetic-induced rat models revealed that the nanofibers had significantly better wound healing features than the control group. The smooth and fine nanofiber with porous web-like structure in the nano range was validated by SEM data. The developed nanofiber had a good mechanical strength and the required biodegradability. Water retention studies demonstrated that the nanofibers could keep the wound site moist and critical for effective wound healing. Studies of in-vitro drug release have shown that nanofiber has sustained release of AT for a long time period. Anti-microbial tests revealed that the produced nanofibers have good anti-microbial action against both Gram-positive and Gram-negative bacteria (*P. aeruginosa* and *S. aureus*). In-vitro cell line studies show less cytotoxicity of nanofibers, and cell migration assay results reveal the increased migration of cells to fill the gap. Furthermore, the histology study revealed that nanofiber has wound-healing characteristics. As a result of the findings, it can be concluded that the produced nanofiber formulation has effective properties of diabetic wound healing in rats and can be processed further for clinical testing. This formulation’s improved scale-up ability paved the way for future commercialization, indicating that it could be a useful tool for diabetic wound healing. Long-term investigations in this direction can be continued.

## Figures and Tables

**Figure 1 pharmaceutics-14-00273-f001:**
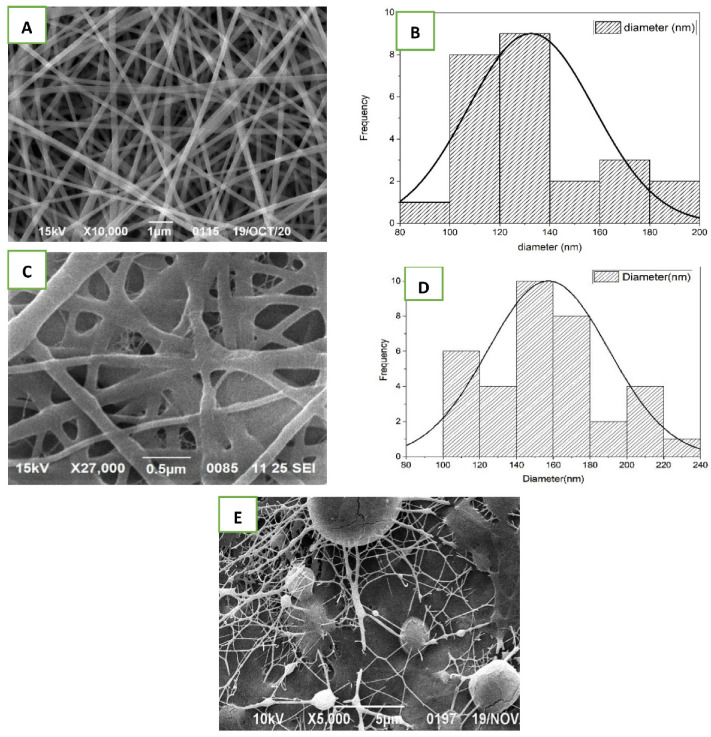
SEM images of nanofibrous scaffolds and their diameters. (**A**,**B**) AT-PVA-SA-SF (non-crosslinked); (**C**,**D**) AT-PVA-SA-SF (cross-linked); (**E**) silk-fibroin nanofibrous scaffold.

**Figure 2 pharmaceutics-14-00273-f002:**
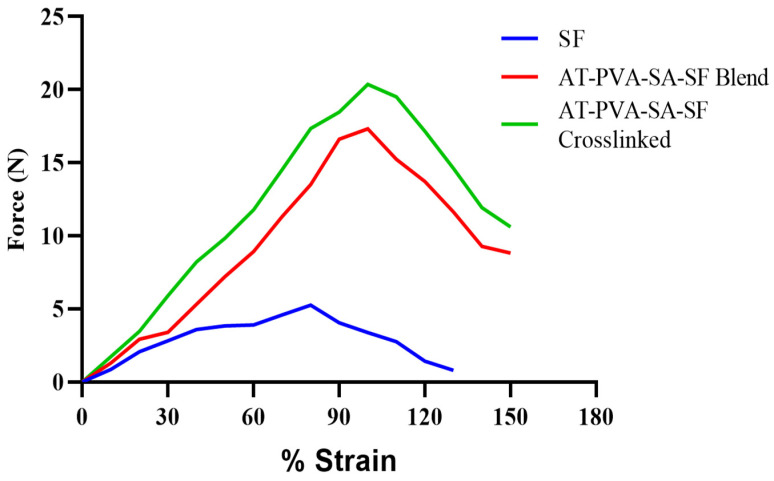
Tensile strength curve of nanofibrous formulations.

**Figure 3 pharmaceutics-14-00273-f003:**
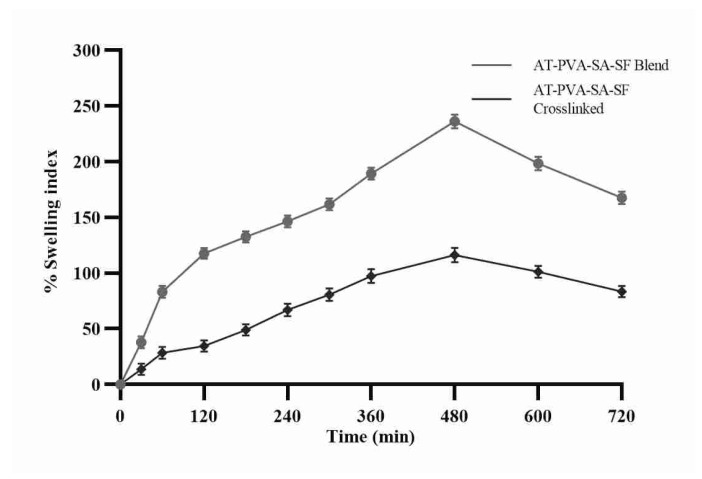
Swelling index graph of nanofibrous formulations.

**Figure 4 pharmaceutics-14-00273-f004:**
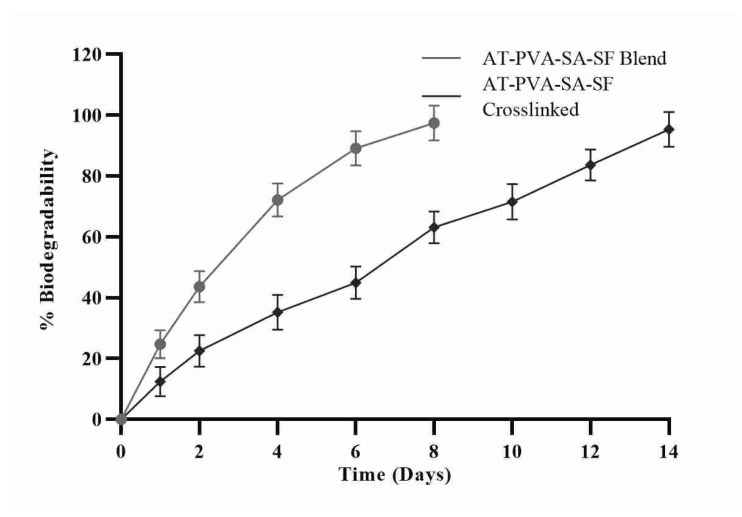
% Biodegradability of nanofibrous formulations.

**Figure 5 pharmaceutics-14-00273-f005:**
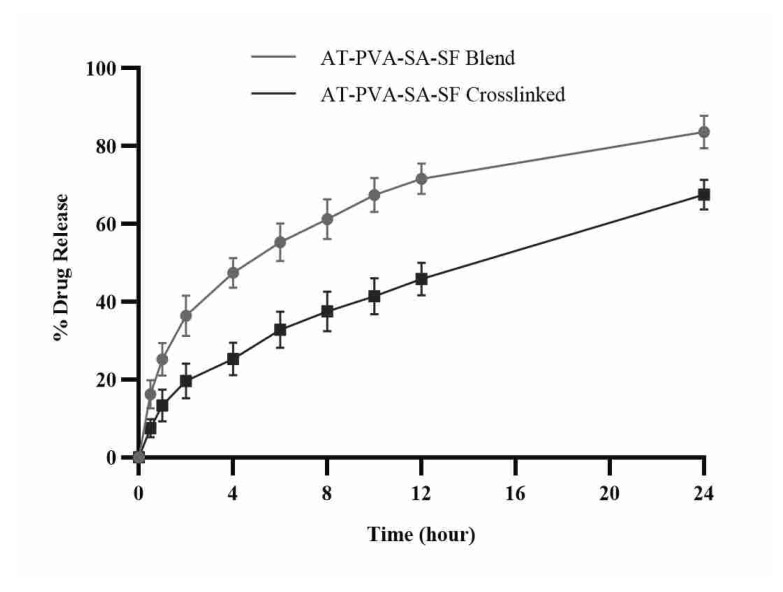
In vitro drug release of AT-loaded nanofibers.

**Figure 6 pharmaceutics-14-00273-f006:**
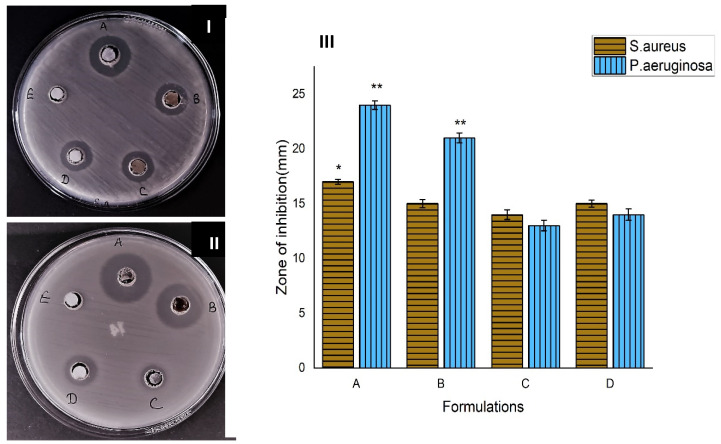
Zone of inhibition of nanofibers against bacteria (**I**) *S. aureus* and (**II**) *P. aeruginosa*. (**A**)-Non-cross-linked AT-PVA-SA-NF; (**B**)-cross-linked AT-PVA-SA-NF; (**C**)-SF-NF; (**D**)-positive control and (**E**)-negative control and (**III**) Bar diagram of agar disc diffusion assay results. The error bars indicate the mean ± S.D. (*n* = 3) and * indicates significant *p* = 0.05; ** indicates significant *p* = 0.01.

**Figure 7 pharmaceutics-14-00273-f007:**
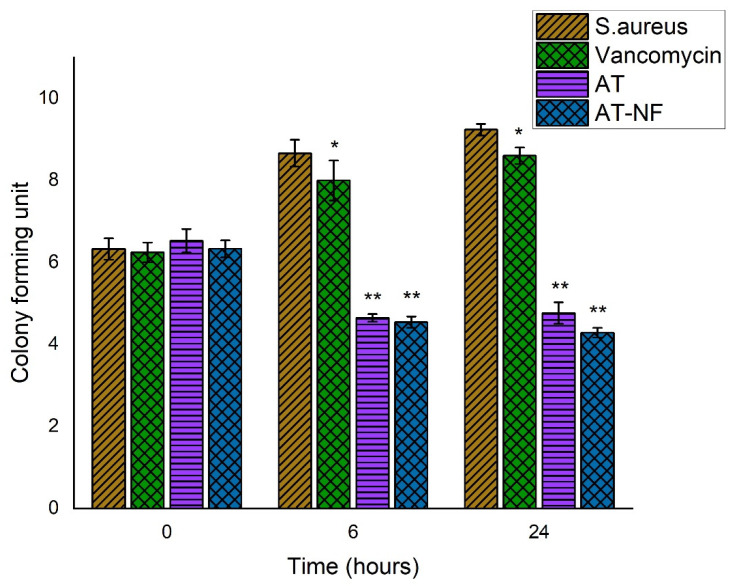
Bar diagram of time-kill assay results against *Staphylococcus aureus*. The error bars indicate the mean ± S.D. (*n* = 3). *p* = 0.05 (* represents significant value) and *p* = 0.01 (** represents highly significant value) represent the significant differences between control, drug and the formulations corresponding to bacteria.

**Figure 8 pharmaceutics-14-00273-f008:**
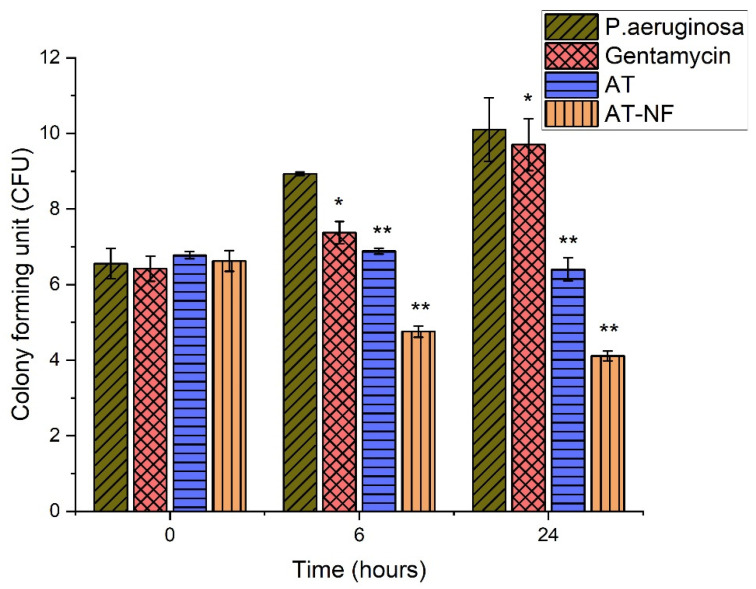
Bar diagram of time-kill assay results against *Pseudomonas aeruginosa*. The error bars indicate the mean ± S.D. (*n* = 3). *p* = 0.05 (* represents significant value) and *p* = 0.01 (** represents highly significant value) represent the significant differences between control, drug and the formulations corresponding to bacteria.

**Figure 9 pharmaceutics-14-00273-f009:**
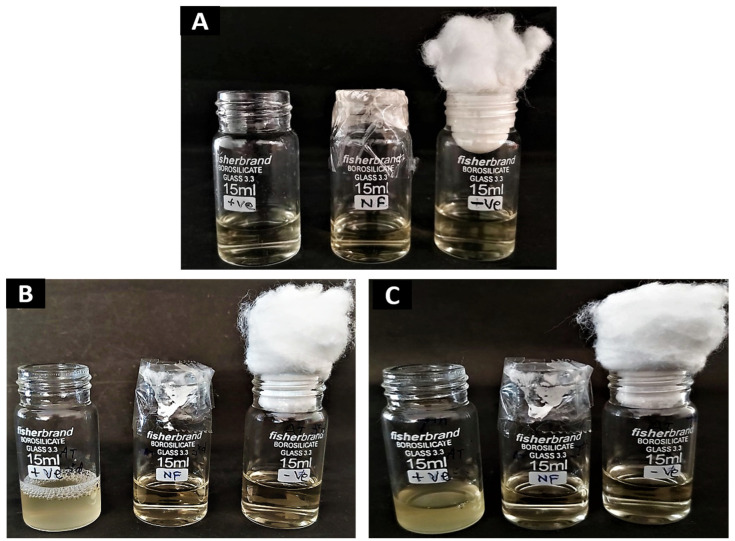
Microbial penetration assay -first open vial as a positive control; second vial capped with AT nanofiber and third vial plugged with cotton as a negative control at (**A**) 0 day, (**B**) 7 days and (**C**) 14 days.

**Figure 10 pharmaceutics-14-00273-f010:**
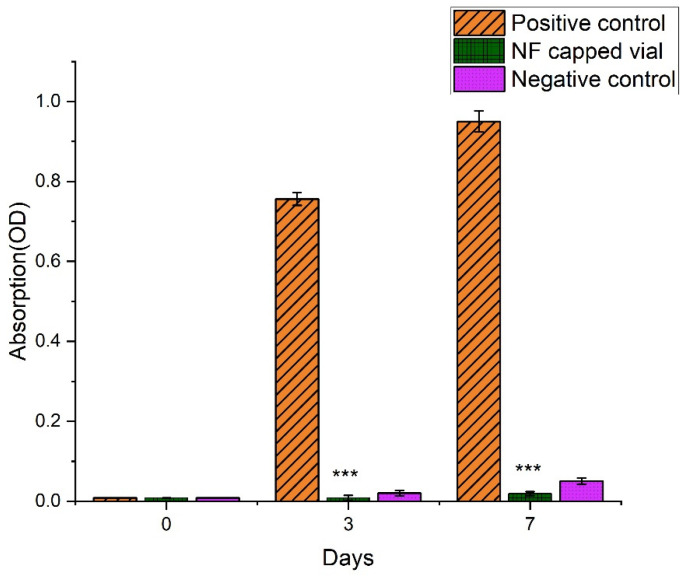
Bar diagram of microbial penetration assay of AT-loaded nanofibers (*** represents highly significant value, *p* = 0.001) against positive and negative control.

**Figure 11 pharmaceutics-14-00273-f011:**
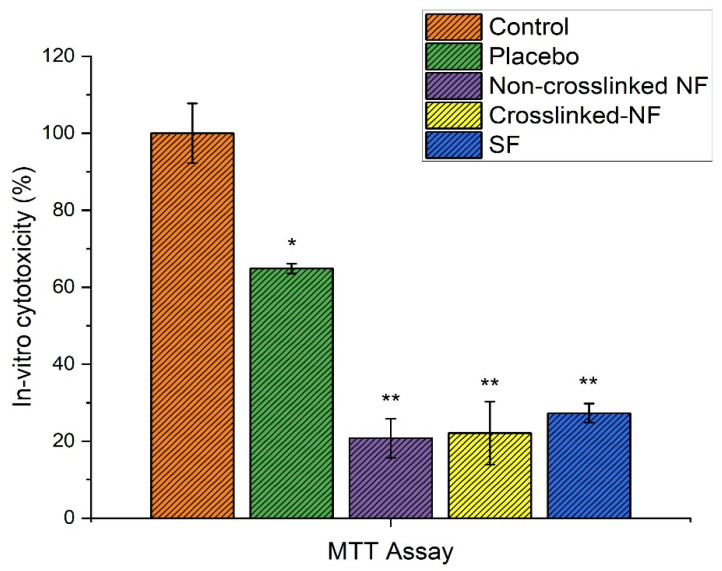
Percentage graph of in vitro cytotoxicity of all the formulations through MTT assay (* represents significant value *p* = 0.05 and ** represents highly significant value *p* = 0.01).

**Figure 12 pharmaceutics-14-00273-f012:**
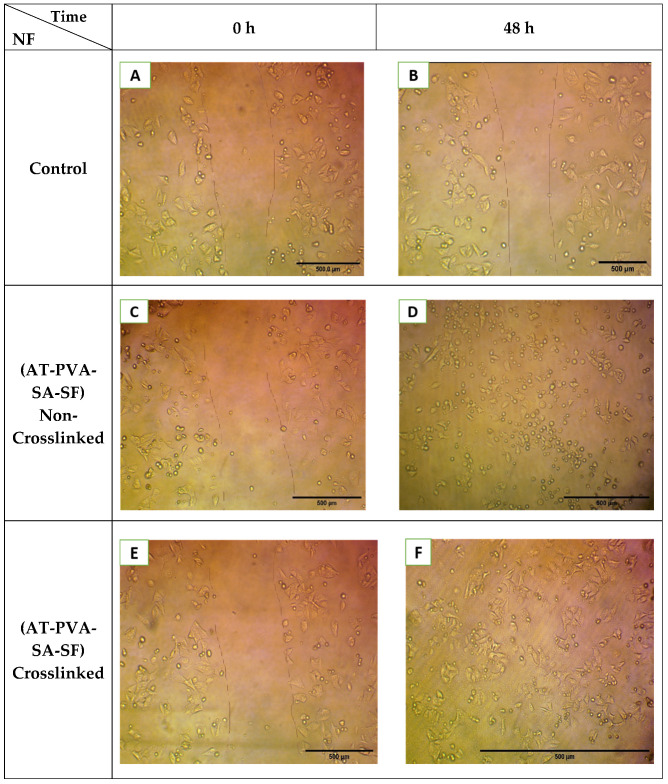
Images of in vitro scratch or cell migration assay of formulations—control ((**A**)-0 h & (**B**)-48 h), non-crosslinked ((**C**)-0 h & (**D**)-48 h) and crosslinked ((**E**)-0 h & (**F**)-48 h).

**Figure 13 pharmaceutics-14-00273-f013:**
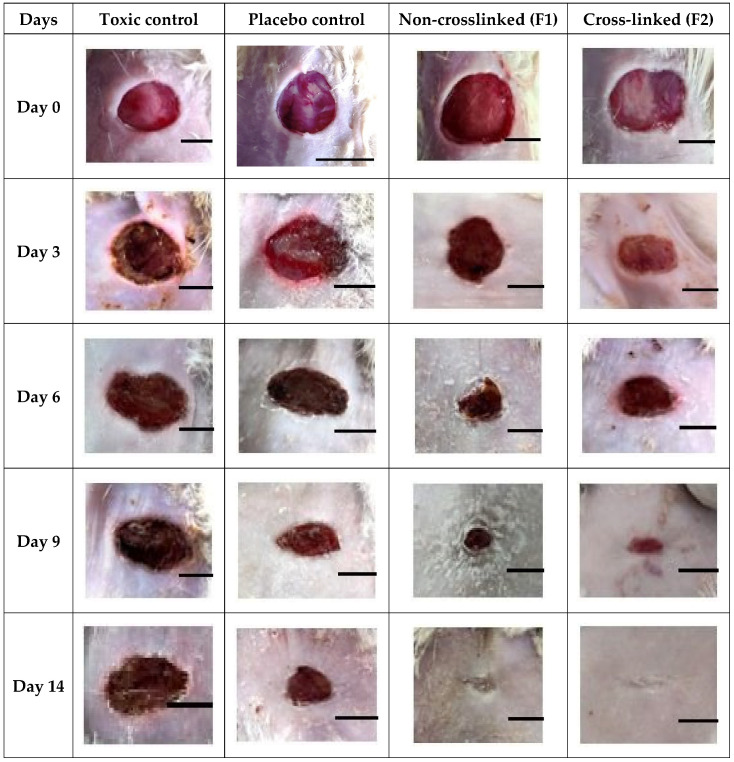
Photographic images of the healing of wound in diabetic-induced rats on 0, 3rd, 6th, 9th and 14th day after treatment of animals with TC (Toxic control), NC (normal control), F1 (non-crosslinked, AT-PVA-SA-SF) and F2 (cross-linked AT-PVA-SA-SF) (Scale bar-4 mm).

**Figure 14 pharmaceutics-14-00273-f014:**
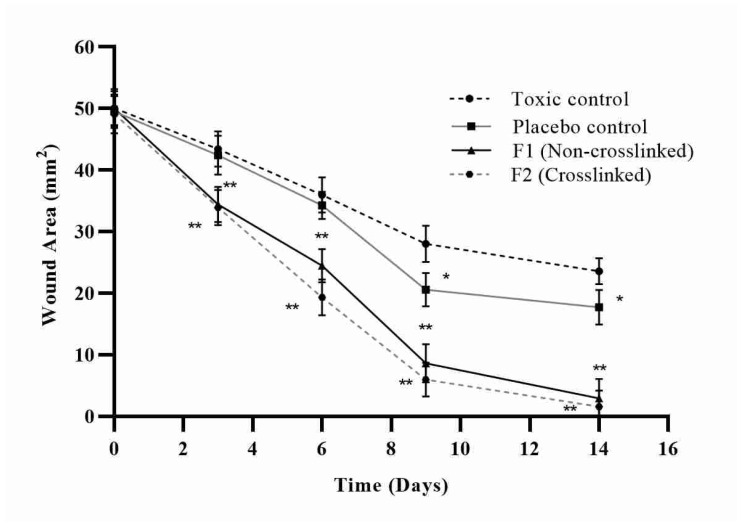
Wound healing for toxic control, normal control, F1 (non-crosslinked) and F2 (crosslinked) (Mean ± S.D. * represents significant value *p* = 0.05 and ** represents highly significant value *p* = 0.01).

**Figure 15 pharmaceutics-14-00273-f015:**
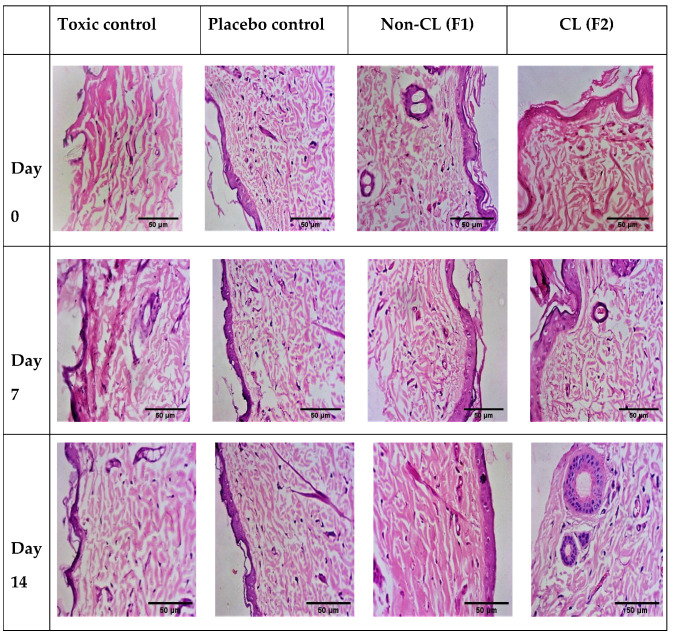
Histopathological images of normal control, toxic control, F1 (non-crosslinked) and F2 (crosslinked).

**Table 1 pharmaceutics-14-00273-t001:** Tensile strength of nanofibrous formulations.

S. No.	Formulations	Strength (MPa)
1.	AT-SF-NF	6.04 ± 0.56
2.	AT-PVA-SA-SF (Non-crosslinked)	17.73 ± 1.23
3.	AT-PVA-SA-SF (Crosslinked)	20.65 ± 1.79

**Table 2 pharmaceutics-14-00273-t002:** Mean blood sugar level of animals before and after inducing diabetes.

Groups	Mean Sugar Level (mg/dL)		
	0 Day	3rd Day	14th Day
	Before Inducing Diabetes	After Inducing Diabetes	
**NC**	90 ± 4.54	170 ± 4.68	110 ± 2.63
**TC**	100 ± 4.28	300 ± 4.57	130 ± 4.38
**F1**	95 ± 3.87	250 ± 3.65	120 ± 3.22
**F2**	100 ± 4.99	280 ± 4.62	135 ± 4.51

**Table 3 pharmaceutics-14-00273-t003:** Mean body weight (gm) of animals before and after inducing diabetes.

Groups	Mean Body Weight (gm)		
	0 Day	3rd Day	14th Day
	Before Inducing Diabetes	After Inducing Diabetes	
**NC**	130 ± 2.08	100 ± 2.10	125 ± 2.19
**TC**	130 ± 2.17	105 ± 2.31	134 ± 2.67
**F1**	145 ± 3.02	110 ± 3.08	140 ± 3.48
**F2**	150 ± 2.98	130 ± 2.88	140 ± 2.43

## Data Availability

Not applicable.

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
