# Peer review of "Multifunctional Biomimetic Nanofibrous Scaffold Loaded with Asiaticoside for Rapid Diabetic Wound Healing"

_pharmaceutics, 2022, doi:10.3390/pharmaceutics14020273_

Round 1
Reviewer 1 Report
Dear Editor,
The manuscript entitled Multifunctional Biomimetic Nanofibrous Scaffolds Loaded with Asiaticoside for Rapid Diabetic Wound Healing is interesting and address an important clinical topic. Chronic wounds as a consequence of diabetes mellitus are a significant clinical problem that can involve serious health issues. The study is focus on the preparation and characterization of a novel electrospun mats, as well as their effect on bacteria, HaCaT cells and in vivo using male rats.
The study is well designed and performed, although the manuscript presents several mistakes and the lack of results description and discussion.
The introduction is well organized and allows to put the reader in context of the current state-of-the-art. In addition, the materials and methods are described in detail, with only several points that could be clarified.
On the other hand, the results section shows several mistakes. The number of figures is not correct, the nomenclature of the nanofibers is not clear and some results are not discussed. To highlight some of them:
- Figure 1E is not described in figure legend.
- Figure 2 must be Table 1.
- Figure 3 is not mentioned in results section.
- Figure 5 shows the biodegradation of non-cross-linked nanofibers until day 8, although in text is described that disintegrate on day 5 (no data is shown on day 5 in the graph).
- Figure 8 and 9 do not contain statistics.
I recommend the authors to revise the manuscript carefully and correct the mistakes, not only the main ones I mentioned.
The experimental work is well done, so I consider that the authors can work on the manuscript to improve the results section. In addition, some paragraphs of the Results and Discussion section could be re-write to clarify the main findings.
I will be able to review the manuscript again after the changes proposed. However, in the correct form is not possible to consider it for publication.
Author Response
Responses of Reviewer 1 comments
Point 1- Figure 1E is not described in figure legend.
Response 1- Figure 1E was described in figure legend as silk fibroin NF but due to typing mistake it was typed as D in the place of E. It has been corrected in the figure legend.
Point 2- Figure 2 must be Table 1
Response 2- Yes, the figure 2 was the representation of table 1 but the numbering of figure was mistaken from figure caption. Now, It has been corrected.
Point 3- Figure 3 is not mentioned in results section.
Response 3- According to the manuscript, figure 3 was mentioned in the results section but as we discussed in response 2, the numbering was mistaken. The numbering of all the figures has been updated.
Point 4- Figure 5 shows the biodegradation of non-cross-linked nanofibers until day 8, although in text is described that disintegrate on day 5 (no data is shown on day 5 in the graph).
Response 4- As shown in figure 5 the biodegradation of non-cross-linked nanofiber was until day 8 but in the text by mistake, it was written until day 5. Now, it has been corrected in the text to day 8.
Point 5- Figure 8 and 9 do not contain statistics.
Response 5- The statistics was performed by one way analysis of variance and unpaired, two sided student t-test and the differences were found significant at 95% confidence interval.
Note: As the reviewer recommended, we have corrected all the corrections. Fig 13, Day 3 (placebo) image was repeated in Day 0 (placebo). We have corrected and updated in the manuscript.
Reviewer 2 Report
The authors have developed a multifunctional biomimetic nanofibrous scaffold loaded with asiaticoside for rapid diabetic wound healing. However, there are many parts that confuse us.
- The author declared that all animals weight was 150-200 g in Part 2.12 (Line 272), 250-300 g in Part 2.12.1 (Line 278), while in Part 3.9 the weight of animals was 90-100 before inducing diabetes (Line 493) and 130-150 (Line 494). In addition, animal number of five in each group is not sufficient.
- There are lots of abbreviations in the legends we do not know, such as NF in Figure 1; AT, SF, PVA, SA, AT-PVA-SA-SF in Figure 3, and things like that. In addition, the Figure 2 is missing, Figure 3-Figure * were wrong numbered. And the number of strength in Table 1 should repeat more than three times.
- In Figure 7, A and B should be changed to (Ι) and (П) (Line 416-418). In addition, Figure 7 should be combined with Figure 8.
- From the MTT test in Figure 13, the cell viability of scaffold was less than 70%, however, the author claimed that these scaffolds were biocompatible and non-toxic to HaCaT cells. And the textdescription of Y-axis should be changed to “Cell viability (%)”.
- Cell state and cell density in Figure 14 were non-ideal.
- The qaulity of presentation should be highly improved, and the format of Figures should be highly improved.
Author Response
Responses for Reviewer 2 comments
Point 1- The author declared that all animals weight was 150-200 g in Part 2.12 (Line 272), 250-300 g in Part 2.12.1 (Line 278), while in Part 3.9 the weight of animals was 90-100 before inducing diabetes (Line 493) and 130-150 (Line 494). In addition, animal number of five in each group is not sufficient.
Response 1- We were working on different manuscripts at the same time. So, there was a manual error had been occurred during writing this manuscript. Actually, we took 100-150 g of animals for the entire study. In Part 2.12 (Line 272), Part 2.12 (Line 278), the actual weight was 100-150 g instead of 150-200 g and 250-300 g, respectively. In Part 3.9, we mentioned the mean blood sugar level (mg/dl) in the table (Line 493), not the average body weight. In line 494, the data indicates the weight of the animals before inducing diabetes and the weight was 130-150 g which lies in between the range of the animal weight we took for the entire study, i.e., 100-150 g.
We conducted the in-vivo study by taking a sufficient number of animals, i.e., n=6, but after inducing diabetes, we saw a casualty of animals in few groups, due to which we immediately started the studies with n=5. During the studies, no casualty was observed thereafter and the study was conducted successfully.
Point 2- There are lots of abbreviations in the legends we do not know, such as NF in Figure 1; AT, SF, PVA, SA, AT-PVA-SA-SF in Figure 3, and things like that. In addition, the Figure 2 is missing, Figure 3-Figure * were wrong numbered. And the number of strength in Table 1 should repeat more than three times.
Response 2- All the abbreviations are updated in the manuscript at appropriate places in the manuscript. There was an error in numbering in Figure legends, figure 2 is mentioned as Figure 3. So, we have updated correctly the numbering in Figure legends like Figure 2, Figure 3, and so on.
The tensile strength of nanofibers was taken in triplicate and the strength (MPa) was also given in the mean, but we forgot to mention ±S.D. The ±S.D. has been updated in Table 1 (Line 360).
Point 3- In Figure 7, A and B should be changed to (Ι) and (П) (Line 416-418). In addition, Figure 7 should be combined with Figure 8.
Response 3- Figure 7, A and B has been changed to (I) and (II) and combined with Figure 8 as suggested by the reviewer.
Point 4- From the MTT test in Figure 13, the cell viability of scaffold was less than 70%, however, the author claimed that these scaffolds were biocompatible and non-toxic to HaCaT cells. And the text description of Y-axis should be changed to “Cell viability (%)”.
Response 4- As per the Para 2.11.2. (Line 246) we conducted an MTT study for in-vitro cytotoxicity. The results of the same are discussed in Para 3.7. (Line 449) and in Figure 13. In Figure 13, the result indicates that in-vitro cell cytotoxicity of scaffolds was less than 20%.
Point 5- Cell state and cell density in Figure 14 were non-ideal.
Response 5- The cell line studies were performed in HaCaT cell line . The revival of the cell line was very difficult and tedious due to the nature of cell line. However, we have put our best effort to revive the HaCaT cell and performed the studies. We obtained the maximum cell density of cells after 48 hrs as shown in Fig 14.
Point 6- The quality of the presentation should be highly improved, and the format of Figures should be highly improved.
Response 6- We have seriously considered all the comments received from reviwers, and improved /corrected the overall presentation of the manuscript. We have addressed every comment received from the reviewer and corrected the manuscript accordingly.
Round 2
Reviewer 1 Report
Dear Editor,
As previously commented in first round of review, the study is well designed and performed, although the manuscript presents several mistakes and the lack of results description and discussion.
The authors have improved the manuscript according my previous comments, although the manuscript needs better description of the results and figures. A lot of concepts and nomenclature are still unclear, including the names of the nanofibres. Only minor changes have been performed to the manuscript and I think the whole manuscript must be revised by authors to make it clear and correct important mistakes.
I can highlight some of the most important issues, but I really think that the authors should work hard to revise the manuscript in detail:
- All figures legends should describe properly the nanofibers types (using the same name for all figures), e.g. AT-PVA-SA-SF, and always in the same order. It is written in some parts as AT-SF-PVA-SA.
- The figures 1A and 1C are not cited in the text.
- Table 1 is not cited in the text.
- Statistical analyses are included in the figure legend of figure 6, 7, 8. However, the statistical analysis should be also represented in the graphs as asterisks, letters or other methods to describe statistical differences. In addition, the other figures do not contain any detail of statistics.
- Figure 10 is not mentioned in the text. Which results are?
- F1 and F2 of figures 14 and 15 should be defined.
Apart from the mistakes I have already mentioned, I have several comments on the results and discussion:
- In section 3.2 authors described that “The tensile 357 strength of the developed nanofibrous formulation was in the range of 12–18 MPa, which 358 is within the ideal range for cutaneous application.” However, crosslinked nanofibers showed 20.65 MPa. Could the authors explain this sentence?
- The biodegradability results showed that non crosslinked nanofibres disintegrated completely on day 8. Did the crosslinked ones disintegrate completely on day 14?
- In figures 7 and 8, what is the meaning of S. aureus and P. aeruginosa columns? And Vancomycin and Gentamycin?
- Which group does the placebo columns represent?
I recommend the authors to revise the manuscript carefully and correct the mistakes, not only the main ones I mentioned.
I will be able to review the manuscript again after the changes proposed. However, in the correct form is not possible to consider it for publication.
Author Response
Round 2 responses of reviewer 1 comments
Comment 1- All figures legends should describe properly the nanofibers types (using the same name for all figures), e.g. AT-PVA-SA-SF, and always in the same order. It is written in some parts as AT-SF-PVA-SA.
Response 1- The nanofiber types order has been corrected as per the reviewer's suggestion. The corrected order is mentioned properly throughout the text i.e., AT-PVA-SA-SF (line no.132)
Comment 2- The figures 1A and 1C are not cited in the text.
Response 2- Figures 1A and 1C have been cited in the text. (line no. 328, 335)
Comment 3- Table 1 is not cited in the text.
Response 3- Table 1 has been cited in the text. (line no.375)
Comment 4- Statistical analyses are included in the figure legend of figure 6, 7, 8. However, the statistical analysis should be also represented in the graphs as asterisks, letters or other methods to describe statistical differences. In addition, the other figures do not contain any detail of statistics.
Response 4- The statistical analysis has been represented in the graphs as asterisks to describe statistical differences in the figures as per the reviewer's suggestion.
Comment 5- Figure 10 is not mentioned in the text. Which results are?
Response 5- Figure 10 has been mentioned in the text (line no 474) . By mistake, figure 8 was repeated in figure 10. Now it has been corrected /changed and these are the results of microbial penetration assay (fig.9) and the same results are represented in bar diagram (fig.10).
Comment 6- F1 and F2 of figures 14 and 15 should be defined.
Response 6- F1 and F2 of Figures 14 and 15 have been defined. F1 is non-crosslinked nanofibers and F2 is crosslinked nanofibers.
Comment 7- In section 3.2 authors described that “The tensile 357 strength of the developed nanofibrous formulation was in the range of 12–18 MPa, which 358 is within the ideal range for cutaneous application.” However, crosslinked nanofibers showed 20.65 MPa. Could the authors explain this sentence.
Response 7- The tensile strength of human skin is from 1-32 MPa. So the tensile strength of crosslinked nanofibers, i.e., 20.65 MPa is compatible with human skin and helps in skin remodelling. However, in some papers the range is given till 18 MPa. Therefore, we have added both the references (39,40) in the manuscript.
Comment 8- The biodegradability results showed that non crosslinked nanofibres disintegrated completely on day 8. Did the crosslinked ones disintegrate completely on day 14?
Response 8- The non-crosslinked nanofiber disintegrated completely on day 8, while cross-linked nanofiber disintegrate almost completely (95%) because as we found some residues (5%) were left during the study.
Comment 9- In figures 7 and 8, what is the meaning of S. aureus and P. aeruginosa columns? And Vancomycin and Gentamycin.
Response 9- In figure 7 and 8, S. aureus and P. aeruginosa columns represents negative control as these are bacterial growth without any treatment. Vancomycin and Gentamycin column represents positive control because these are the drugs that inhibit bacterial growth.
Comment 10- Which group does the placebo columns represent?
Response 10- The placebo column has not been taken for this study as placebo nanofiber does not contain any drug. Fig 7 and 8 represents the time-kill assay of drug-loaded nanofiber formulation that shows the growth inhibition at different time intervals.
Note: In addition to the specific comments by the reviewer, we have also checked the whole manuscript including figures and Tables, and corrected all minor mistakes/corrections, grammar, spelling mistakes, etc.. we have also checked language editing /grammar correction using Grammarly premium version.
Reviewer 2 Report
We suggest that author revise the manuscript carefully.
The in-vitro cell cytotoxicity of scaffolds was slightly higher than 20% in Figure 11.
Author Response
Round 2 response of Reviewer 2 comment
Comment- The in-vitro cell cytotoxicity of scaffolds was slightly higher than 20% in Figure 11.
Response- The exact value of scaffold cytotoxicity was 22. We have written just approximate value of cell viability, i.e., 80% and cell toxicity, i.e., 20%.
Round 3
Reviewer 1 Report
The authors revised the manuscript according to my comments. I recommend the editor to accept to manuscript in the present form.
Author Response
Dear Reviewer,
Please note that we have revised the manuscript of round 3 revision and you also recommended the editor accept it in the present form.
Kindly do the needful for the same and convey your recommendation to the editor again